triact package for R: analyzing the lying behavior of cows from accelerometer data

Simmler Michael 1 michael.simmler@agroscope.admin.ch
http://orcid.org/0000-0002-7028-7823 Brouwers Stijn P. 2
1 Digital Production, Agroscope , Ettenhausen , Switzerland
2 Centre for Proper Housing of Ruminants and Pigs, Federal Food Safety and Veterinary Office (FSVO), Agroscope , Ettenhausen , Switzerland
Sunny Armando
Electronic publication date: 2024 Feb 28
Publication date: 2024
Volume: 12
Electronic Location ID: e17036
Received 2023 Dec 12; Accepted 2024 Feb 9
Copyright: © 2024 Simmler and Brouwers
Copyright year: 2024
Copyright holder: Simmler and Brouwers
License: This is an open access article distributed under the terms of the Creative Commons Attribution License, which permits unrestricted use, distribution, reproduction and adaptation in any medium and for any purpose provided that it is properly attributed. For attribution, the original author(s), title, publication source (PeerJ) and either DOI or URL of the article must be cited.
License URL: https://creativecommons.org/licenses/by/4.0/

Keywords: Activity, Cattle, Animal welfare, Sensor

Funding: Federal Food Safety and Veterinary Office on behalf of the Swiss Government 2.21.01 This publication benefitted from financial support provided by the Federal Food Safety and Veterinary Office on behalf of the Swiss government (grant number: 2.21.01). The funders had no role in study design, data collection and analysis, decision to publish, or preparation of the manuscript.

==============================
Accelerometers are sensors proven to be useful to analyze the lying behavior of cows. For reasons of algorithm transparency and control, researchers often prefer to use their own data analysis scripts rather than proprietary software. We developed the triact R package that assists animal scientists in analyzing the lying behavior of cows from raw data recorded with a triaxial accelerometer (manufacturer agnostic) attached to a hind leg. In a user-friendly workflow, triact allows the determination of common measures for lying behavior including total lying duration, the number of lying bouts, and the mean duration of lying bouts. Further capabilities are the description of lying laterality and the calculation of proxies for the level of physical activity of the cow. In this publication we describe the functionality of triact and the rationales behind the implemented algorithms. The triact R package is developed as an open-source project and freely available via the CRAN repository.

Introduction

Lying behavior is of great interest regarding the welfare, productivity, and health of cows. On average, lactating dairy cows lie down for a total of around 8 to 13 h per day, divided into typically between 9 and 11 lying bouts averaging 60 to 100 min in length (Tucker et al., 2021). There is clear evidence for lying being a high-priority behavior for cows. Cows sacrifice other behaviors, such as feeding and social behaviors, to lie down and show rebound lying behavior after periods of forced standing (Tucker et al., 2021). Adequate lying times are therefore considered important for animal welfare and decreased lying times and other alterations of lying behavior are used as indicators of compromised welfare. Consequently, lying behavior is a particular focus of attention in animal welfare assessments of housing systems (Beaver, Weary & von Keyserlingk, 2021; Mee & Boyle, 2020), management options such as stocking rate (Krawczel & Lee, 2019), and aspects of stall construction such as the type of resting surface or cubicle design (McPherson & Vasseur, 2020). Furthermore, changes in lying behavior are a key behavioral response recorded in studies on heat stress in cows (Galán et al., 2018; Ji et al., 2020), a topic with increasing importance in the near future. Altered lying behavior can also result from, and therefore indicate, injury and illness (Dittrich, Gertz & Krieter, 2019; Mainau et al., 2022). Increased lying time and increased number of lying bouts might be used as indicators of lameness (Sadiq et al., 2017; Whay & Shearer, 2017), and lying laterality, i.e., the side the cow is lying on, might be indicative of mastitis (Medrano-Galarza et al., 2012).

The most common measures for lying behavior include the total duration in lying position during a given period (typically a day), the number of lying bouts in that period, and the mean duration of lying bouts (Tucker et al., 2021). These measures can be reliably determined using a triaxial accelerometer attached to a hind leg, replacing time-consuming direct visual observations or analysis of video recordings by a human observer. Accelerometers measure proper acceleration, i.e., the acceleration an object experiences relative to free fall. An accelerometer at rest will therefore measure 1 g in skywards direction owing to the mechanical force from the ground, preventing it from free fall. We will henceforth refer to this static acceleration as gravitational acceleration, although, to be precise, it is due to the opposing reaction force, rather than the gravitational force itself. An accelerometer attached to a cow measures the (vector) sum of this static gravitational acceleration and the dynamic body acceleration from the cow’s movements. The gravitational component is used for the analysis of lying behavior. It is derived from the raw signal either through filtering in the time domain by calculating central tendency over a sliding window (e.g., median filter) or through filtering in the frequency domain by applying a low-pass filter (e.g., Butterworth low-pass filter). An accelerometer attached to the hind leg of a standing cow measures approximately 1 g of gravitational acceleration on the upward axis as the leg is more or less perpendicular to the surface of the earth (the upward axis is defined as directed along the leg towards the torso). When the cow lies down, it descents onto its carpal joints and places the hind leg of the intended lying side behind the opposite forelimb and underneath the body. The body is then lowered to rest on the lower hind leg, thigh, and abdomen (see e.g. Lidfors, 1989). During lying, the lower hind leg of the lying side is thus comparably fixed in place, more or less perpendicular to the sagittal plane and parallel to the ground surface. The other hind leg is also more or less horizontally placed but has more freedom of movement. It is bent and extended to varying degree in direction of the cow’s head, or extended away from the torso (Fig. 1). When transitioning from standing to lying, the reference frame of the accelerometer attached to the hind leg is rotated relative to the direction of gravity. An apparent change in direction of the gravity component of the acceleration is observed, shifting from the upward axis to the other axes. This shift is exploited to determine lying vs. standing posture and lying laterality.

Figure 1 Natural lying positions of cows. Reproduced from Schnitzer (1971) with permission of KTBL.

Commercial systems such as IceTag and IceQube (IceRobotics, South Queensferry, Scotland) employ a leg-attached triaxial accelerometer, automatically apply proprietary algorithms to the raw accelerometer data, and return the processed lying behavior data. However, for reasons of algorithm transparency and better control on settings such as sampling rate, many researchers prefer to use generic accelerometers (e.g., by MSR Electronics GmbH, Seuzach, Switzerland) and use their own data analysis scripts written in a programming language such as R (R Core Team, 2023).

Here, we present the triact R package, an open-source software that assists the analysis of lying behavior of cows from triaxial accelerometer data (manufacturer-agnostic). Starting from raw data files, the triact package provides methods to calculate total lying duration, number of lying bouts, and mean lying bout duration, all broken down by the lying side (lying laterality). In addition, it allows the periods in standing posture to be described using the same measures, therefore enabling equal study of standing and lying as requested by Tucker et al. (2021) in their review. Finally, triact allows calculating proxies for the level of physical activity of the cows. As an open-source R package, triact encourages science that can be reproduced, better understood, and verified.

Design considerations

The targeted users of the triact R package are animal scientists. Many design decisions were based on several years of experience with in-house R scripts for analyzing cow lying behavior used at Agroscope, the Swiss center of excellence for agricultural research. The finalization of the R package benefited from the study by Brouwers et al. (2023), from which all data presented in this publication originate. Care was taken to keep the software lightweight with respect to dependencies. It directly depends on four packages that are not part of the core distribution of R, namely R6 (Chang, 2022), data.table (Dowle & Srinivasan, 2021), lubridate (Grolemund & Wickham, 2011), and checkmate (Lang, 2017). These dependencies are very lightweight themselves, causing only three secondary dependencies to further packages (two from lubridate, one from checkmate). Non-compulsorily, triact furthermore depends on the R packages signal (Signal Developers, 2013) and tibble (Müller & Wickham, 2019) for some non-default options. Like R itself, all dependencies are available for free and published under an open-source license.

To be able to sequentially add analyses as columns to the tabular data, and to avoid creating data copies, we opted for object-oriented programming using the R6 system and employed data.table to operate on the data by reference. This is essential to keep the memory footprint as small as possible—as accelerometer data is recorded at high sampling frequency (>1 Hz), datasets can become large. In addition, we can leverage data.table’s optimizations for performance, including its internal parallelization to use multiple CPU threads for common operations. While data hidden to the user (encapsulated in an R6 object) is stored in the data.table variant of R’s data.frame, the type of table returned by user-exposed methods can be set via a global option to either data.frame (the default), data.table, or tibble. The latter option allows triact to nicely play within the popular tidyverse R dialect (Wickham et al., 2019). We programmed defensively; using functionality of the checkmate R package, all input of user-exposed methods is checked and meaningful error messages are instantly generated if appropriate.

Functionality

In the following we introduce the functionally of the triact R package v.0.3.0 available via the Comprehensive R Archive Network (CRAN; https://cran.r-project.org/web/packages/triact). We focus on background and rationale and leave the executable examples to the vignette distributed with the package.

Data to be analyzed with triact should be recorded with a triaxial accelerometer attached to the hind leg on the outward facing side of the metatarsus. It must be represented on three Cartesian axes (XYZ) in units of g (=9.81 m s−2). Typical accelerometer sampling frequencies used with triact are between 1 and 20 Hz. Depending on the manufacturer and the orientation of the accelerometer on the leg, its XYZ axes might point in different directions. To avoid confusion, the triact package renames the axes according to body relative directions (Fig. 2A). When the cow is standing, the forward and upward axes form a plane parallel to the sagittal plane, while the right axis is perpendicular to it (Fig. 2B). On importing data, the user is questioned on how to map the accelerometer’s XYZ axes to these body relative axes.

Figure 2 Definition of directions.

(A) Body relative directions. (B) The directions as used in triact, corresponding to body relative directions when the cow is standing. Depicted is what the situation looks like when the accelerometer is mounted on the outside of the left hind leg (or on the inside of the right hind leg). Silhouette of the cow from Vecteezy.com. Silhouette of the cow leg reproduced from Hendriks et al. (2020a) with permission of Elsevier.

Importing data

The Triact R6 class encapsulates both data and all core functionality (Fig. 3). After creating a new instance of the Triact class, acceleration data of one or several cows is imported into the object with a $load_… method. Calling $load_files() processes the raw accelerometer files and imports them into the R6 object. Given the method’s arguments are appropriately specified, delimiter-separated text files with different characteristics can be imported, e.g., different separators, file header length, data-time formats, and more. Alternatively, users can process raw data on their own and import the concatenated data as a data.frame-like table using the $load_table() method. However, this requires the user to exactly prepare the data as requested (column names, data types, etc.). Imported raw data and added analyses (see next paragraph) can be accessed between any step of the workflow via the $data field.

Figure 3 Schematic overview of the workflow enabled by methods and fields of the Triact R6 class.

Prefixes in method names indicate their role in the workflow. The code in this figure is executable if you follow the flow as indicated in the flowchart. However, it is intended to provide an overview of the workflow rather than to teach the use of the R package. For the latter, we have written a vignette that is distributed with the R package (https://cran.r-project.org/web/packages/triact/vignettes/introduction.html).

Detecting lying and standing posture

Calling $add_… methods triggers analyses of lying behavior and the calculation of proxies for the level of physical activity. These analyses are obtained for each time point of the acceleration data and added in a new column to the tabular data in the Triact object.

The $add_lying() method performs the classification into lying and standing. Standing is defined as “not lying” and includes walking and other activities in upright posture. The two postures lying and standing are thus mutually exclusive and collectively exhaustive. The simple rule-based algorithm is composed of three steps and uses only information from the upward axis (Fig. 4). In the first step, the upward acceleration is filtered to obtain the gravity component of the signal, by default by using a moving median filter with a window size of 10 s (Fig 4B). In the second step, a threshold (crit_lie; Fig. 4B) is used to classify the filtered acceleration values into standing (>0.5 g) and lying (<0.5 g). The default threshold of 0.5 g corresponds to a sensor tilt or angulation of the leg of about arccos(0.5/1 g) ≙ 60° (Fig. 5). Finally, in the third step, lying bouts shorter than 30 s are “removed” by reclassifying them as standing. This step aims to remove falsely detected short lying bouts (false positives).

Figure 4 Determination of standing and lying posture and lying laterality.

(A) Results on standing and lying posture and lying laterality (left L, right R) as determined with triact. (B) Acceleration as measured on the upward axis (up) with median filtered data and critical lying threshold indicated as dashed horizontal line. (C) Acceleration as measured on the right axis with threshold for determining lying side indicated as dashed horizontal line. Data obtained at 20 Hz sampling frequency with a triaxial accelerometer mounted on the left hind leg of a dairy cow.

Figure 5 Observed gravitational acceleration as a function of the angle of the hind leg.

(A) The angle between the upward axis (blue arrow) and the vertical as a function of the static gravitational acceleration on the upward axis (formula given in figure). The dashed line indicates the threshold used in triact to discriminate between standing and lying posture. (B) Trigonometric visualization of the calculation of the critical angle of 60° from the threshold that is crit_lie = 0.5 g. The gravitational acceleration is directed skywards as the accelerometer measures proper acceleration (see Introduction). Silhouette of the cow leg reproduced from Hendriks et al. (2020a) with permission of Elsevier.

The first two steps of the algorithm are sufficient to reliably detect lying bouts of ~2 min and longer. In fact, virtually perfect accuracy in detecting these bouts is expected, with uncertainties similar to direct observation by a human observer (a few seconds at most). The total daily lying duration can therefore be expected to be highly accurate, as the contribution of the rare and less reliably detectable very short lying bouts (a minutes or less) to the total lying duration is typically negligible (Tucker et al., 2021). Detecting such short lying bouts with limited reliability can however significantly compromise accuracy of the daily number of lying bouts (and consequently also the number of standing bouts). Challenging is the avoidance of short false positive lying periods corresponding to the cow standing but lifting the hind leg on which the accelerometer is attached toward the horizontal position for several seconds to up to half a minute or more (Fig. S1; Videos S1 and S2). Typically, this occurs in self-grooming behaviors. Because true short lying periods of less than a minute are uncommon when the cow is not forced to do so, it is reasonable to set a minimal duration for a lying period and correct accordingly (third step in our algorithm). The IceTag sensors with proprietary algorithm apparently suffer from the same challenges, do not (sufficiently) address this issue, and manual exclusion of short lying bouts post hoc is common when results of these sensors are used in scientific context (e.g., Hendriks et al., 2020b; Tolkamp et al., 2010). In a study with 28 lactating dairy cows, Kok et al. (2015) reported the shortest true lying bout to be 33 s long and that correcting the IceTag sensor output using 33 s as a minimum duration maximized accuracy. As this agrees well with our experience that true lying bouts shorter than 30 s and lifting the hind leg in standing posture longer than 30 s are both very rare (not systematically studied), this seems to be a threshold that adequately balances the tradeoff between true positive and false negative lying bouts. However, although very rare, lying bouts can be much shorter than the threshold—on the pasture we observed lying bouts as short as ~8 s (Fig. S2; Video S4). Very short standing bouts do occur as well, typically to change lying side (Fig. S3; Videos S5 and S6). Because of physical constraints of how the cow can place the hind legs while lying, false standing bouts in actually lying cows are not expected and have never been observed by us when using the default parameter of triact and acceleration data with sampling frequency of ≥1 Hz. Therefore, it is no surprise that the IceTag sensors with their proprietary algorithms are also reported very reliable when it comes to short standing bouts (Tolkamp et al., 2010).

For users of the triact R package analyzing data sampled at recommended frequencies of ≥1 Hz, we expect good results when using the default parameters of the $add_lying() method. However, we encourage users to experiment with the many parameters in order to increase understanding, particularly in potential validation studies or for “off-label” use with animals other than cows. Besides the moving median filter (default), a bidirectional (zero-lag) Butterworth low-pass filter can be selected to filter for the gravity component in the first step of the algorithm (parameter: filter_method; see Winter (2009) for an explanation of zero-lag Butterworth filter). In the second step, the threshold for binary classification can be specified (crit_lie). In the last step, the minimum duration of lying bouts can be adjusted (minimum_duration_lying). A minimum duration of standing bouts can be specified as well. However, this is not recommended (see above) and not used by default. The filtering methods available for the first step can additionally be fine-tuned if desired. Using median filter (the default), the window size in seconds (window_size) can be adjusted. For the Butterworth filter, both the order (default is first order) and the cutoff frequency (default is 0.1 Hz) can be defined by the user. The default cutoff of 0.1 Hz we found suitable for low-pass filtering to obtain the gravity component is in a similar range as the 0.3 Hz used for high-pass filtering to obtain the body acceleration in studies on dairy cow behaviors by Riaboff et al. (2019) and Smith et al. (2016). Cutoff frequencies between 0.1 and 0.5 Hz are also commonly used in the literature to separate static gravitational from dynamic body acceleration in studies on human posture and movements (Bayat, Pomplun & Tran, 2014; Foerster & Fahrenberg, 2000; Veltink et al., 1996).

Detecting lying laterality

The $add_side() method performs the determination of lying laterality (left or right) based on the information of the right axis (Figs. 2, 4C). For each lying bout detected with $add_lying(), one and the same lying side is assigned for the entire duration of the lying bout. This is justified because change of lying side occurs via a short standing bout, which consequently splits the lying bout in two lying bouts. Furthermore, short standing bouts are reliably detected, even when switching side by standing with only the hind legs fully extended but the front legs resting on the carpal joints (Fig. S3; Video S5). The simple one-step algorithm determines whether the majority of the acceleration values measured during that bout are above (left lying side) or below (right lying side) a threshold (crit_left; Fig. 4C). This is equivalent to comparing the median of the acceleration values with the threshold. The threshold depends on which hind leg the accelerometer was mounted on. It is by default 0.5 g in case of the left hind leg, and −0.5 g in case of the right hind leg. When working with this default value, the user must thus specify which hind leg was used (parameter: left_leg). Alternatively, the threshold can be freely adjusted (crit_left).

Assuming the cow would lay down the leg with the accelerometer in the coronal plane and perfectly horizontal, one would expect that the right axis will detect the gravity component of around ±1 g, with a positive mathematical sign (along its direction) when lying on the left side and a negative mathematical sign (opposing its direction) when lying on the right side. Thus, under this (miss) assumption a threshold of 0 would be a logical choice. However, the above assumptions are poor in situations where the cow is not lying on the side where the accelerometer is attached. Compared to the leg on which the cow is lying, the other hind leg has more freedom of movement and can be placed bent to different degree in the direction of the cow’s head (Fig. 1). This can result in the gravity component partially to nearly completely loading on the forward axis (and not the right axis) resulting in values on the right axis approaching 0. It is therefore necessary to shift the threshold more towards the values expected for the situation with the accelerometer on the lying side, i.e., if the accelerometer is on the left leg towards positive (therefore 0.5), if on the right leg towards negative (therefore −0.5). Figure 4, S2 and S3 exemplarily show that when the cow lies on the left side where the accelerometer was mounted, values tend to be around 1 quite consistently, while when it lies on the right side, the central tendency of values can be quite different from closer to −1 (Fig. 4) to close to around 0 (Fig. S3).

Calculating proxies for physical activity

The $add_activity() method allows calculating proxies for the physical activity level based on the information of all three accelerometer axes. By default, the L2 norm of the dynamic body acceleration (DBA) vector is calculated. The corresponding L1 norm is optionally available. Also, the L1 and L2 norms of the jerk vector can be calculated (Table 1). During lying bouts, all activity values are “adjusted” to 0, i.e., periods when cows are lying are considered as “inactive” by definition (Fig. 6).

Table 1 Proxies for the level of physical activity available in triact.

Proxy	Dynamic measure	Norm	Unit	
L1DBA
(aka ODBA)	DBA	L1	g	
L2DBA
(aka VeODBA)	DBA	L2	g	
L1Jerk	Jerk	L1	g s−1	
L2Jerk	Jerk	L2	g s−1	
Note:

The L1 norm of the dynamic body acceleration vector (L1DBA), the L2 norm of the dynamic body acceleration vector (L2DBA), the L1 norm of the jerk vector (L1Jerk), and the L2 norm of the jerk vector (L2Jerk).

Figure 6 Level of physical activity of a cow as determined with triact.

L2DBA represents the L2 norm of the dynamic body acceleration. “Adjusted” refers to setting the value for the activity level to 0 for the lying periods, which are by definition considered as the “inactive” periods. The underlying acceleration data is shown in Fig. 3 (apart from the forward axis). Data obtained at 20 Hz sampling frequency with a triaxial accelerometer mounted on the left hind leg of a dairy cow.

DBA is obtained by subtracting the static gravity component from the measured acceleration. The static component is obtained by filtering as described above for the lying behavior (same defaults, same options). Jerk, the change in acceleration and therefore its first derivative, is approximated through the backward finite difference method, i.e., difference in acceleration to one time point back divided by time interval (sampling interval respectively). For a vectorial property v→ (DBA or jerk) at time point ti, the L1 norm is the sum of absolute values in each axis and the L2 norm is the square root of the sum of the squared value in each axis:

L1 norm

||v→(ti)||=|vfwd(ti)|+|vup(ti)|+|vright(ti)|

L2 norm

||v→(ti)||2= vfwd(ti)2+vup(ti)2+vright(ti)2

Both norms aggregate the jerk or DBA vector at time point ti into a single non-negative number. Because the L2 norm calculates the magnitude of the vector in Euclidian space, the resulting measures are meaningful in terms of Newtonian physics (whereas the corresponding L1 norms are not). Also, in contrast to the L1 norm, L2 is rotation invariant and thus not affected by accelerometer orientation. We therefore advocate for using the L2 norm although the L1 norm is also commonly used in this context. In practice, the two norms are highly correlated both for jerk and DBA (see Table 2). The choice between the two might therefore be a conceptual rather than a practical one.

Table 2 Pearson correlation coefficients for proxies for the level of physical activity.

	L1DBA	L2DBA	L1Jerk	L2Jerk	
L1DBA	1.000				
L2DBA	0.994	1.000			
L1Jerk	0.797	0.799	1.000		
L2Jerk	0.797	0.803	0.994	1.000	
Note:

Correlations determined using a total of ~1,500 h of acceleration data recorded at 20 Hz sampling frequency with a triaxial accelerometer mounted on the left hind leg of 48 different dairy cows during standing (all activities in upright posture; data from Brouwers et al. (2023)). L1DBA, L1 norm of the dynamic body acceleration vector; L2DBA, L2 norm of the dynamic body acceleration vector; L1Jerk, the L1 norm of the jerk vector; L2Jerk, L2 norm of the jerk vector.

L1 and L2 norms of DBA are popular in studies on wildlife, where they are commonly known as “overall dynamic body acceleration” (ODBA) and “vector dynamic body acceleration” (VeDBA), respectively (Wilson et al., 2020). Beyond interpretation as an operationally defined “activity level,” they are commonly interpreted as proxy for movement-based energy expenditure (Gleiss, Wilson & Shepard, 2011; Halsey, Shepard & Wilson, 2011; Wilson et al., 2006, 2020). For several species, empirical relationships to measures such as rate of oxygen consumption and heart rate were established (Halsey et al., 2009; Miwa et al., 2015). Miwa et al. (2015) found ODBA (L1 norm of DBA) to be a good proxy for estimating energy expenditure of grazing farm animals (cows and goats), given appropriate calibration and considering body mass. The ODBA and VeDBA metrics are generally intended to be based on acceleration measured around the center of mass of the animal (Halsey, Shepard & Wilson, 2011) and not at a hind leg as in triact. However, because acceleration at the leg is strongly related to locomotion, for grazing animals we expect correlation with energy expenditure for physical activity, similar to when acceleration is measured closer to the center of mass. Generally, reports of application of DBA metrics with farm animals are scarce, maybe because in this field of research proprietary metrics such as the “Motion Index” are more common. However, this “Motion Index” as returned from IceTag and IceQube sensors (also hind leg mounted) seems to be defined along the same line as the L1 norm of DBA (“measure of the animal’s activity which considers the absolute value of the 3-D acceleration”, www.icerobotics.com accessed 27.11.22).

Jerk as measure for the general physical activity level seems to be less used than the DBA-based metrics. However, many studies might not refer to it as jerk but rather as differences in acceleration, corresponding to the finite difference approximation of jerk. In Switzerland, the L1 norm of the jerk vector, although not referred to as such, was frequently used as proxy for physical activity or locomotion activity in dairy cows (Gómez et al., 2022; Lutz et al., 2019; Weigele et al., 2018). By approximating jerk, the change in acceleration, we remove the influence of the static gravity component as the latter is not associated with jerk (hence static). However, movements of the animal body will always be associated with some jerk because acceleration does not suddenly switch on or off but ramps up and down more or less smoothly. Therefore, periods with higher absolute values of DBA, e.g., during walking with the repeated acceleration and deceleration during each gait cycle, will be associated with higher absolute values of jerk. Thus, when used across different behaviors and complex movements, jerk is correlated with body relative acceleration. For example, for ~1,500 h of accelerometer data collected by Brouwers et al. (2023) from a hind leg of several dairy cows during standing (all activities in upright posture), the correlations of L1 and L2 norms of jerk with L1 and L2 norms of DBA were r ≈ 0.8 (Pearson; Table 2). Finally, an important note on the apparent physical activity during lying bouts: By default, activity measures during lying bouts are set to a value of 0. This default behavior of $add_activity() can be suppressed by setting the parameter adjust = FALSE. However, then returned non-zero physical activity levels during the lying bouts should be interpreted with caution. When determined using an accelerometer on one hind leg, the measured level of physical activity suffers from a significant bias depending on the lying side. The hind leg on which the cow is lying is comparatively physically restricted, while the other hind leg has more freedom of movement (see introduction). Thus, when the accelerometer is placed on the leg underneath the lying cow, a lower level of activity is measured than if the accelerometer were placed on the opposite leg.

Summarizing lying behavior and activity

The results previously added to the Triact object with the $add_… methods can be summarized using the $summarize_bouts() and the $summarize_intervals() methods. The returned tables are ready to be subjected to downstream statistical analyses (not part of the triact package). With $summarize_bouts(), a summary is created for the individual lying and standing bouts, with duration, mean activity, lying side (for a lying bout), and more. With $summarize_intervals(), the summary is obtained per regular intervals (e.g., a day or an hour). Examples of summary measures are the total duration of lying, the duration of lying on left or right side, the number of lying bouts, the mean duration of lying bouts, and the mean activity. For measures such as the number of lying bouts or mean lying bout duration, a weighted mean is calculated with the weights being the proportion of the individual bout overlapping with the respective interval. For example, a bout completely in the interval contributes with weight 1 vs. a bout covered by the interval only by 30% contributes with weight 0.3 (and with 0.7 to the adjacent interval). This weighting ensures that the totals are not falsely inflated. For both $summarize_… methods, we strictly suppress output (return NA) that depends on ill-defined information of the first and the last bouts (e.g., duration), which are only partly covered by the data. However, the user can consciously override this behavior and force a summary in any case (parameter calc_for_incomplete)—first and last bouts in the data are thereby simply assumed complete.

Extracting posture transitions

Using $extract_liedown() and $extract_standup(), the raw acceleration data (and added analyses) of the posture transitions, i.e., lying-to-standing and standing-to-lying, can be extracted. The data between user-specified times before and after the exact moment of posture transition as detected by triact are returned (parameters sec_before, sec_after). Among other things, this can be useful in developing algorithms able to characterize these posture transitions. For example, Brouwers et al. (2023) used the $extract_… methods to extract regions of interest as a first step in a machine learning–based workflow to detect atypical behaviors performed during lying down and standing up movement patterns.

Off-label use

The triact R package has been developed to analyze the lying behavior of adult cows. The basic principle, that gravitational acceleration can be used to estimate sensor tilt, is generally valid and the simple rule-based algorithms may give the impression that they are also applicable to calves or other animal species (e.g., other ruminants or equids). However, the details of the implementation of the algorithms and the default values of their parameters, e.g., critical values in thresholding steps, are tailored to adult cows. Therefore, satisfactory results when used with animals other than adult cows can not per se be expected and a verification of the functionality for the specific case is necessary. The values of the algorithm parameters may need to be adjusted. A thorough consideration of the behavioral repertoire and the lying behavior of your study animal compared to adult cows may be helpful in this process. The intermediate results of individual steps in triact should be visually inspected when searching appropriate values for parameters (see the package documentation on how to preserve intermediate data).

Conclusions and outlook

The R package triact assists animal scientists in determining common measures to characterize the lying behavior of cows from raw data recorded with a triaxial accelerometer mounted on a hind leg. As open-source project, it warranties full algorithmic transparency. Depending upon suggestions of the research community, we will add new functionality to triact or develop companion R packages. This further development may include functionality for visualizing results. In terms of technical improvements, we plan to implement out-of-memory computation, to remove the prohibitive memory requirements when working with large amounts of high sampling frequency (>1 Hz) accelerometer data. The triact package is developed as an open-source project and welcomes contributors.

Supplemental Information

Supplemental Information 1 Supplementary Figures.

We would like to thank Lorenz Gygax for his contribution to in-house R scripts that led to the development of the triact R package. We are grateful to our work colleagues from the Centre for Proper Housing of Ruminants and Pigs (Agroscope) for the feedback we received on early versions of the R package and to Nadja El Benni and Ralph Stoop for feedback on the manuscript.

Additional Information and Declarations

Competing Interests

Author Contributions

Data Availability

The authors declare that they have no competing interests.

Michael Simmler conceived and designed the experiments, performed the experiments, analyzed the data, prepared figures and/or tables, authored or reviewed drafts of the article, and approved the final draft.

Stijn P. Brouwers conceived and designed the experiments, authored or reviewed drafts of the article, and approved the final draft.

The following information was supplied regarding data availability:

The supplementary videos are available at Zenodo: Simmler, M., & Brouwers, S. P. (2023, January 12). Supplementary videos for “triact package for R: Analyzing the lying behavior of cows from accelerometer data”. Zenodo. https://doi.org/10.5281/zenodo.7528564.

The latest release of the triact R package including vignette and example dataset is available at CRAN: https://cran.r-project.org/web/packages/triact/.

The development version of the package is available at GitHub and Zenodo:

- https://github.com/agroscope-ch/triact.

- michaelsimmler. (2024). agroscope-ch/triact: triact v0.3.0 (v0.3.0). Zenodo. https://doi.org/10.5281/zenodo.10612926.

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
