# Peer review of "triact package for R: analyzing the lying behavior of cows from accelerometer data"

_PeerJ, doi:10.7717/peerj.17036_

## Round 0.1 · original submission · Minor Revisions

Dear Authors,

The consensus from both reviewers underscores the significance of this R package in facilitating robust analyses. Having thoroughly tested the package, they affirm its functionality. Nevertheless, some minor corrections are advised before the manuscript can be considered ready for publication.

Best Regards,

Armando Sunny

·

Basic reporting

I welcome this R-package, because it will allow accelerometer-based research on cow behaviour to become transparent and comparable. I found the paper well-written and easy to follow. Below a few comments.

Discussion: The code was developed based on data from cows, and I fully acknowledge that the title, abstract and introduction also mentions cows. However, I am certain that readers/researchers will ask whether this code is valid for calves or other species, too. Therefore, I encourage the authors to add a paragraph to the Discussion on the limitations of the application of the code, because animals of different size and/or species may not move and behave in ways that allow your code to produce correct results. It would be beneficial to add at least the range of cow body weight which your code was developed for.

l. 170-171: Please provide justification, e.g. a reference, for your choice of 30s as a threshold for false lying bouts.

l. 191: Referring to “our experience” does not convince me, can you please provide some results or provide a reference, which shows what the shortest duration of a true lying bout is?

l. 225-228 Detecting lying laterality. According to l. 135-136, the software requires the accelerometer to be attached to a hind leg (i.e. no side mentioned), with the sensor on the outside of the leg. Will it not be necessary to inform the software which leg the sensor is actually attached to, in order to obtain derive the true lying side when detecting laterality? Your figure 3 (myT$add_side(left_ leg==TRUE)) seems to say that the default side is the left leg. Does this mean, that if the sensor is mounted on the right leg of the cow, the user needs to change the code? If yes, I think this needs to be described clearly in the text. Okay, this is explained further below (l. 235-238). Consider moving it up for emphasis.

l. 261: Consider replacing “adjusted” by “set” – while omitting the quotation marks, because you effectively do set it to zero, which is a constant. With “adjustment” the reader will think the value can be variable according to the adjustment.

Table 1 and 2: Tables should be stand-alone, therefore an explanation to describe what the proxies are, is needed.

Experimental design

no comments.

Validity of the findings

I have not reviewed code formally before, therefore the following comment may not be relevant.
If I were to use the package described, I would want to know how well if performs, and for instance see a test of how well it works. Did you compare how well the code output corresponds with a gold standard, e.g. human observation of cow behavior? It might be sufficient with a short description and a reference to that study.

·

Basic reporting

The work entitled "triact package for R: Analyzing the lying behavior of cows from accelerometer data" presents merit and offers animal scientists an interesting, useful, agnostic, and user-friendly tool to analyze data from accelerometers in relation to the lying behavior of cows.
In addition to the well-written text with many resources for understanding the functions and how to use them, the authors took care to keep the package lightweight and with few dependencies. Furthermore, they used the object-oriented paradigm (OOP) to build the package, which aligns very well with modern software development practices. The work is quite reproducible and after testing all the functions exposed in the paper and in the vignette, I managed to obtain the same results. Furthermore, the package's source code can be accessed in its entirety on github, which makes accessing and understanding it even clearer.
Therefore, I recommend publishing the paper in question without any concerns.

Experimental design

In addition to the well-written text with many resources for understanding the functions and how to use them, the authors took care to keep the package lightweight and with few dependencies. Furthermore, they used the object-oriented paradigm (OOP) to build the package, which aligns very well with modern software development practices.

Validity of the findings

The work is quite reproducible and after testing all the functions exposed in the paper and in the vignette, I managed to obtain the same results. Furthermore, the package's source code can be accessed in its entirety on github, which makes accessing and understanding it even clearer.
Therefore, I recommend publishing the paper in question without any concerns.

---

## Round 0.2 · accepted · Accept

Dear Esteemed Authors,

It brings me great pleasure to inform you that following the necessary revisions, your manuscript stands poised for publication in PeerJ.

Best Regards,

Armando Sunny

·

Basic reporting

The authors have responded and amended the manuscript satisfactorily, and the explanation for off-label use is very informative.
I have no further comments.

Experimental design

No further comments

Validity of the findings

No further comments

Additional comments

No further comments